# *Yersinia pestis* Plasminogen Activator

**DOI:** 10.3390/biom10111554

**Published:** 2020-11-14

**Authors:** Florent Sebbane, Vladimir N. Uversky, Andrey P. Anisimov

**Affiliations:** 1Université de Lille, Inserm, CNRS, CHU Lille, Institut Pasteur de Lille, U1019—UMR9017—CIIL—Center for Infection and Immunity of Lille, F-59000 Lille, France; 2Department of Molecular Medicine and USF Health Byrd Alzheimer’s Research Institute, Morsani College of Medicine, University of South Florida, Tampa, FL 33612, USA; 3Laboratory of New Methods in Biology, Federal Research Center “Pushchino Scientific Center for Biological Research of the Russian Academy of Sciences”, Institute for Biological Instrumentation of the Russian Academy of Sciences, 142290 Pushchino, Moscow Region, Russia; 4Laboratory for Plague Microbiology, State Research Center for Applied Microbiology and Biotechnology, Especially Dangerous Infections Department, 142279 Obolensk, Moscow Region, Russia

**Keywords:** *Yersinia pestis*, plasminogen activator, omptin, pathogenicity factor, pathogenesis, plague

## Abstract

The Gram-negative bacterium *Yersinia pestis* causes plague, a fatal flea-borne anthropozoonosis, which can progress to aerosol-transmitted pneumonia. *Y. pestis* overcomes the innate immunity of its host thanks to many pathogenicity factors, including plasminogen activator, Pla. This factor is a broad-spectrum outer membrane protease also acting as adhesin and invasin. *Y. pestis* uses Pla adhesion and proteolytic capacity to manipulate the fibrinolytic cascade and immune system to produce bacteremia necessary for pathogen transmission via fleabite or aerosols. Because of microevolution, *Y. pestis* invasiveness has increased significantly after a single amino-acid substitution (I259T) in Pla of one of the oldest *Y. pestis* phylogenetic groups. This mutation caused a better ability to activate plasminogen. In paradox with its fibrinolytic activity, Pla cleaves and inactivates the tissue factor pathway inhibitor (TFPI), a key inhibitor of the coagulation cascade. This function in the plague remains enigmatic. Pla (or *pla*) had been used as a specific marker of *Y. pestis*, but its solitary detection is no longer valid as this gene is present in other species of *Enterobacteriaceae*. Though recovering hosts generate anti-Pla antibodies, Pla is not a good subunit vaccine. However, its deletion increases the safety of attenuated *Y. pestis* strains, providing a means to generate a safe live plague vaccine.

## 1. Discovery, History, Genetic Control, Biosynthesis Conditions, Isolation, Purification, and Physicochemical Properties of Plasminogen Activator

### 1.1. The Discovery

Plague is a fatal disease transmitted by fleas, but it can also be contracted through inhalation of contaminated aerosols and the ingestion of contaminated food. It is in 1894 that Alexandre Yersin isolated its causal agent, the Gram-negative bacterium named *Yersinia pestis* [1,2]. In 1936, R.R. Madison revealed that *Y. pestis* displays a fibrinolytic activity (i.e., the capability to break down the product of blood coagulation, a fibrin clot) [3]. However, this activity varies with the source of fibrin. In fact, a comparison of the fibrinolytic activity using fibrin from several mammal species indicated that the activity against human and guinea pig fibrins was about six times lower than that against rat fibrin. Furthermore, gopher, cat, rabbit, cow, and monkey fibrins were even more resistant to lysis. Lastly, no lysis of horse, ram, and pig fibrins was noted. Eight years after the discovery of fibrinolysin by R.R. Madison, E. Jawetz, and K.F. Meyer discovered that *Y. pestis* has also the ability to coagulate plasma [4]. However, this coagulase activity has been demonstrated in the rabbit plasma clotting assay [5], but could not be observed using human, mouse, or rat plasma [6]. Almost 20 years after Jawetz and Meyer’s discovery, Domaradsky reported that the fibrinoyltic and the coagulase activities of *Y. pestis* are interrelated [7,8]. He also indicated that these traits differentiate *Y. pestis* from its most recent ancestor species, *Yersinia pseudotuberculosis*, an enteropathogenic bacterium causing mild bowel disease in humans, which is not transmitted by fleas or aerosols. In the same period of time as I.V. Domaradsky, R.R. Brubaker et al. reported that fibrinoyltic and coagulase activities are linked to the production of a bacteriocin (named “pesticin”), which is active against *Y. pseudotuberculosis* but not *Y. pestis* [9,10]. It turned out that genes responsible for fibrinoytic/coagulase activity and bacteriocin production are all harbored by a small plasmid named pPst (pPla, pPCP1, or pYP) [11,12] that contains at least three genes including *pla*. This latter was rapidly found to encode the enzyme (plasminogen activator, Pla) responsible for both fibrinolytic and plasma-coagulase activities of *Y. pestis* [13,14] and had adhesive properties [15]. Lastly, we must mention that some strains, including one clinical isolate recovered from the blood of a fatal human plague case, was shown to contain the pPst dimer [16]. This dimeric form of the plasmid is stably inherited and also stably coexists with the monomeric form of the pPst. More recently, the simultaneous presence of two different pPst plasmids, one with and one without the *pla* locus, was found in most of the sequenced genomes from the end of the second-plague-pandemic burials, but not in the *Y. pestis* strains circulating during the Black Death peak. It has been suggested that the proliferation of Pla-depleted strains could, among other reasons, contribute to the disappearance of the second plague pandemic in eighteenth-century Europe [17]. Furthermore, we also have to emphasize here that not all the *Y. pestis* strains harbor the pPst plasmid and, accordingly, any of its genes, including *pla*. 

### 1.2. Pla, Location, Processing, and Conformation

Pla is a member of the omptin family, which is composed of outer membrane proteins classified as bacterial aspartate proteases [18,19,20,21]. Hence, Pla shares high homology with four surface enterobacterial proteases: PgtE (*Salmonella enterica*), OmpT (*Escherichia coli*), OmpP (*E. coli*), and SopA (*Shigella flexneri*) [22,23,24,25]. Pla is composed of 312 amino acid residues (no cysteine residues) and includes the N-terminal signal sequence (i.e., the mature protein is 292 amino acid residue-long) [13]. Upon removal of the signal peptide (during protein exportation), an α-Pla (37 kDa) protein is formed. Being an integral outer membrane protease (omptin), the plasminogen activator Pla from *Y. pestis* is characterized by a highly conserved β-barrel structure spanning the membrane. Figure 1 shows that Pla forms a narrow β-barrel with an elliptical cross-section. This rather extended, ~70 Å long barrel consists of 10 antiparallel β-strands connected by short periplasmic turns and five extracellular loops that define the substrate specificity of this bacterial fibrinolysin [18,19]. The export of α-Pla yields to the production of two other forms, β-Pla (35 kDa) and γ-Pla (31 kDa), which are associated with the outer membrane as well [26,27]. V.V. Kutyrev et al. reported the existence of the σ-variant (23 kDa) of Pla [26]. However, there was no confirmation of this information by other researchers. β-Pla is formed as a result of the autoprocessing of the C-terminal surface loop of the α-Pla molecule at amino acid residue Lys_262_ [28]. The α- and β-forms do not differ in their fibrinolytic activity [28]. Although γ-Pla displays lower molecular weight than α-Pla, it is generally accepted that this form of the protein is an alternative form of α-Pla folding [28]. Indeed, the boiling of samples in the presence of SDS led to the disappearance of γ-Pla [29]. 

Similar to other omptins, Pla is an aspartate protease with the catalytic site residues Asp_84_, Asp_86_, Asp_206_, and His_208_ located at the extracellular opening of the β-barrel. Akin to other aspartate proteases, Pla uses a water molecule for cleavage of target peptides [30]. In fact, in the active Pla molecule, catalytic residues are grouped into Asp_84_/Asp_86_ and Asp_206_/His_208_ couples located at opposite sides of the barrel, with the minimal distance between these active site couples being ~4.7 Å. This distance is between Asp_84_ in one couple and His_208_ in another couple, with the catalytic nucleophile water molecule being located between the carboxyl group of Asp_84_ and the Nɛ2 atom of His_208_ [30].

It was pointed out that in a crystal structure of an active form of Pla, residues Glu_252_–Ser_269_ in loop L5 are missing (this region of mature Pla corresponds to the residues 273–290 in the preprocessed protein, UniProt ID: P17811), likely due to the Pla autocatalytic activity [30]. As sites of the proteolytic attack are preferentially located within flexible regions or intrinsically disordered protein regions (IDPRs) [31,32,33,34], this loop is likely characterized by high structural plasticity. Despite such conformational flexibility, loop L5 seems to play an important role in controlling Pla activity. This notion is supported by an interesting observation that in the ancestral *Y. pestis* lineages Microtus and Angola, this loop L5 carries a single amino acid change Thr259Ile that affects the fibrinolytic activity of Pla [35]. In fact, Pla from the Microtus lineage was more efficient than the wild-type Pla in α_2_-antiplasmin inactivation, suggesting that this ancestral form of Pla evolved into a more efficient plasminogen activator in the pandemic *Y. pestis* lineages [35].

To see if loop 5 of *Y. pestis* is indeed disordered/flexible and to evaluate the intrinsic disorder status of the transmembrane protein Pla, we used predictors of intrinsic disorder to study the intrinsic disorder profile of Pla *pestis* and of several other omptin family outer membrane proteases (PgtE from *S. enterica*, OmpT and OmpR from *E. coli*, and SopA/IcsP from *S. flexneri*) (Figure 2). Although sequence identities of these different protease ranges from 72.44% (between Pla and PgtE) to 39.41% (between Pla and SopA/IcsP) or even 38.76% (between PgtE and SopA/IcsP), their disorder profiles are characterized by remarkable similarity, showing conserved positioning of major disordered and flexible regions. More particularly, Pla (despite being an integral transmembrane protein) is predicted to have several IDPRs (i.e., regions with the disorder scores exceeding the threshold of 0.5) and several flexible regions (i.e., regions possessing disorder scores between 0.2 and 0.5) (Figure 2A). They also show that loop L5 is predicted to be disordered. Furthermore, other extracellular loops (residues 53–57 (L1), 108–116 (L2), 165–185 (L3), and 228–232 (L4)) are also predicted to be disordered by at last one of the computational tools used here. Signal peptide, which is removed during maturation of Pla is also an IDPR. These observations suggest that intrinsic disorder and conformational flexibility are important for the functionality of Pla. Further support to the idea of potential functional importance of disordered and flexible regions of Pla is given by their evolutionary conservation.

### 1.3. Regulation of Pla

It has been initially reported that *pla* is regulated neither at the transcriptional nor at the translational level, suggesting that the temperature dependence of the fibrinolytic activity may be due to temperature-induced modifications or conformational changes in the Pla protein [36]. This was somewhat consistent with the presence of −10 and −35 core promoter elements recognized by sigma factor 70 located upstream of the transcription start sites of *pla*. However, comparative transcriptomic analysis using bacteria cultured *in vitro* in artificial media indicated a modest overexpression of *pla* (two-fold changed) after temperature shift from ≤28 °C to 37 °C [37]. This transcriptional activation was further correlated with a subsequent qRT-PCR study [38] and a comparative proteomic analysis [39], as well as western-blot analysis [40]. However, transcriptome analysis of *Y. pestis* showed no induction of *pla* at 37 °C in human plasma [41]. Furthermore, *pla* expression levels in the liver and lungs of mice sacrificed 4 days after intravenous challenge were about 10-fold lower than those seen in vitro at 37 °C [38]. We must point out though that these results were obtained using a strain of *Y. pestis* lacking the pigmentation locus (*pgm*) which may encode a factor affecting *pla* expression and/or activity [42]. Indeed, WT strain displays far greater fibrinolytic activity than *pgm*-negative strain. In any case, the regulation of *pla* in vivo appears complex. Being one of the genes the most highly expressed in the bubo of moribund rat [43], *pla* was found to be repressed in the lung of mice 48 h after intranasal inoculation [44,45] prior to being over-expressed >10-fold in the lungs of animals displaying a terminal stage of infection [45]. Based on *in vitro* molecular studies [46,47,48] and an investigation of mice infected by the intranasal route, it has been proposed that the in vivo expression pattern of *pla* in infected lungs is driven by the glucose concentration and depends on the transcriptional regulator cyclic AMP (cAMP) receptor protein (CRP). In this model, *Y. pestis* entering the lung encounters an environment, where the concentration of glucose is high enough to maintain an intracellular concentration of cAMP low to avoid CRP activation. Consequently, CRP is unable to bind to a site upstream of *pla* between bp −70 and −52 to activate the *pla* transcription [47]. During the infection, *Y. pestis* consumes the glucose present in the lungs. Therefore, at the terminal stage of infection, the environmental glucose concentration is so low that the intracellular concentration of cAMP is high enough to activate CRP, and thereby promote the CRP-dependent *pla* expression. In conclusion, *pla* is expressed thanks to –70 dependent promoter, whose activity is enhanced in the absence of glucose due to the direct binding of CRP to the Pla promoter. Temperature > 25 °C can induce *pla* according to a mechanism that remains to be elucidated.

### 1.4. Role of Temperature and Lipooligosaccharide (Los) on Pla Activities

The modest transcriptional activation of *pla* during a shift from ≤28 °C to 37 °C alone cannot explain the dramatically higher fibrinolytic activity of the bacterium *Y. pestis* cultured at 37 °C vs. ≤28 °C [36]. Furthermore, this apparent regulation does not find an explanation in the amount of Pla per cell. In fact, comparative proteomic analysis and western-blot analysis showed a 2-fold increase in the Pla production, when bacteria are shifted from ≤28 °C to 37 °C [39,40]. Accordingly, E. Ruback et al., who compared the ability of *Y. pestis* grown at 37 °C or ≤28 °C to activate plasminogen, reported that the higher enzymatic activity of “37 °C" bacteria was accompanied by a 1.6-fold increase in the Pla content in the outer-membrane preparations, and a slight increase in the specific activity of “37 °C” Pla [29]. According to immunoblotting, the β-form of Pla was detected only at 37 °C. In other words, the Pla proteolytic activity (at least autoproteolysis and plasminogen activation) depends on a post-translation event. The role of the interaction of Pla with the outer membrane was incriminated. Indeed, for isolation and purification of Pla, gel filtration of cell membrane extracts using Sephadex G-200 was previously used. In the process of isolation, the preparations retained fibrinolytic but lost their coagulase activity [5]. More recently it was found that the product of *pla*, renatured in the presence of LOS, retains all the studied properties of a partially purified molecule, presumably because the interaction of Pla with LPS affects the structure of the protein catalytic site, generating its correct conformation [18,19].

Although the interaction of the outer membrane with Pla was suspected to play a role in plasminogen activation, differences in presence of O-antigen was rapidly excluded, since *Y. pestis* produces a LOS. However, it is interesting to note that the fibrinolytic activity of Pla, but not its auto-processing, requires the absence of O-side polysaccharide chains synthesized by numerous bacterial species (such as *Yersinia pseudotuberculosis* and *S. enterica*) [40,49,50,51]. In other words, Pla is functional, when it interacts with the LOS naturally produced by the *Y. pestis* cells [51].

Since O-antigen modifications cannot explain the difference of Pla activity at the different growth temperature, temperature-dependent change in the structure of lipid A was suspected for the enzymatic activity of Pla. At 37 °C, *Y. pestis* expresses tetra-acylated LOS, whereas the LOS is mainly hexa-acylated when the bacterium is grown at ≤28 °C, incriminating the role of low acylation of lipid A in Pla activity [51]. On the one hand, the “acylation” idea was supported by the experiments using *E. coli* expressing Pla and producing a rough LPS with lower acylation [40]. On the other hand, Pla activities remained unchanged in strains of *Y. pestis* engineered to produce only low or high lipid A acylation regardless of the growth temperature [51,52,53,54]. 

Lastly, the role of the LOS core in the temperature-dependent activities of Pla was also examined. It was found that the shortening of the core part of the LOS of *Y. pestis* or the LPS of *E. coli* producing Pla negatively impacts Pla activity [40]. Furthermore, gradual shortening of the core was correlated with the decrease in the Pla activity, and variants of the core oligosaccharide containing four or less monosaccharide residues loss the fibrinolytic and coagulase activities of Pla [55]. In other words, the data indicate that the presence of a complete core but not its precise carbohydrate structure is critical for Pla activity. 

A number of prokaryotic and eukaryotic proteins that bind to LPS have conserved three-dimensional structures formed with the participation of two positively charged arginine residues interacting with the negatively charged phosphate groups of lipid A [11]. In the Pla molecule, these arginines are located at positions 138 and 171. The Arg171Glu amino acid substitution significantly reduced the enzymatic activity of the Pla preparation, refolding of which was carried out in the presence of LOS. At the same time, the Arg138Glu replacement or the double Arg138Glu + Arg171Glu replacement led to a complete loss of the Pla ability to activate plasminogen [29]. Variants of Pla with the Arg138Glu orArg171Glu substitutions formed a significantly lower amount of β-Pla compared to the wild-type molecules. The double Arg138Glu + Arg171Glu mutant was not able to form β- and γ-Pla at all, which indicates the necessity of binding to LOS for autoprocessing and correct folding of Pla [29]. Overall, these data led to the idea that the requirement of LOS for Pla activity could represent a means to keep the protease inactive during the translocation from the cytoplasm into the outer membrane [19].

## 2. Pla as a Broad-Spectrum Protease with Adhesion Properties 

### 2.1. The Multiple Substrates Processed by Pla 

#### 2.1.1. The Substrates Related to Hemostasis 

Coagulation and fibrinolysis are complex biological phenomena minutely balanced by the intervention of numerous factors playing either an activator or a repressor role (Figure 3), and where an inhibited target can impede its own inhibitor [56,57,58]. Following trauma or injury, a cascade of enzymatic cleavages causes prothrombin to be cleaved into thrombin, which splits fibrinogen. This results in the release of fibrin, which forms a mesh that traps blood cells and is therefore serves as an essential element in the formation of a blood clot. The fibrin clot can also be degraded by the action of the peptidase named plasmin, which is a product of the degradation of plasminogen by the action of tissue plasminogen activator (tPA) and the urokinase-type plasminogen activator (uPA), with the latter being primarily involved in cell migration and tissue remodeling linked to the inflammation and bacterial infections. The proteolytic activity of tPA and uPA can be balanced by the plasminogen activator inhibitor-1 and -2 (PAI-1 and -2), whereas α2-antiplasmin, α-2-macroglobuline, as well as the thrombin-activatable fibrinolysis inhibitor (TAFI) impede the plasmin activity. PAI (stabilized when complexed with vitronectin [Vn]) is a competitive inhibitor composed of a sequence (“a bait”), whose integration into the catalytic site of uPA or tPA leads to the formation of an irreversible covalent bond between the inhibitor and the enzymes. The α2-antiplasmin (A2AP) is a serine protease inhibitor (serpin) that degrades plasmin. As for the activated TAFI, it decreases plasmin production and fibrinolysis by removing lysine residues from partially degraded fibrin, which reduces the binding of t-PA and plasminogen to the fibrin clot. Interestingly, activated TAFI (TAFIa), as well as TAFI, are subjected to destructive proteolysis by plasmin. It is worth noting that both fibrinolysis and coagulation are linked to an immune response. For instance, PAI-1 participates in immune cell migration, binding to cell surface receptors, and cytokine production.

Thanks to its proteolytic activity, Pla disturbs several factors belonging to both coagulation and fibrinolytic pathways. Notably, Pla seems to induce coagulation by inactivating the tissue factor pathway inhibitor (TFPI), a protease that inhibits the factor (F) VII initiating one of the first steps of blood coagulation (Figure 3) [59]. This induction of coagulation could be further enhanced by the activation by Pla, albeit not very efficiently, of FVII [59]. It is worth noting that no hydrolysis was found towards the FV and FX [59]. 

Paradoxically and in the opposite way to the coagulation activation, Pla induces fibrinolysis. The direct cleavage of the plasminogen by Pla, at the same site with uPA (i.e., R561-V562), was the first mechanism identified that explains how Pla induces fibrinolysis [6,60]. Subsequent studies that have reviewed the different actors in the fibrinolytic cascade sensitive to Pla have revealed that plasmin production is amplified both by activation of uPA [61] and also by the degradation of several fibrinolysis inhibitors such as PAI-1 [30,62,63], A2AP, and TAFI. Pla cleaves PAI-I in the inhibitory “bait”, between R346 and M347, resulting in the inability of the inhibitor to covalently obstruct the catalytic site of uPA [30]. The inactivation of PAI-1 is further accentuated by the hydrolysis of its stabilizer Vn to different pieces by Pla [30,63]. However, efficient degradation of Vn requires the presence of the adhesin Ail that confers binding to Vn [63]. The precise site of cleavage is unknown for A2AP and TAFI, although, Pla is known to cleave TAFI at a Ct region [64]. As aforementioned, plasmin inactivates TAFI and TAFIa, indicating that Pla, directly and indirectly, lowers the potential inhibitory effects of TAFIa on fibrinolysis. In conclusion, Pla acts, directly and indirectly, to produce plasmin from plasminogen. When it acts indirectly, it activates plasminogen activators and degrades the inhibitors of these activators as well as plasmin. Altogether, it is clear that the different actions of Pla lead to high-level production of plasmin. Since plasmin degrades several components of the extracellular matrix in addition to fibrin, it appears reasonable to consider that the fibrinolytic activity of Pla can lead to uncontrolled proteolysis. However, as described later on, this uncontrolled proteolysis does not necessarily play a role or a role of the same importance in every site colonized by *Y. pestis*. 

Furthermore, we would like to emphasize here that the ability of Pla to degrade some components of the fibrinolytic activity in vitro is not necessarily observed in vivo. Besides, when a substrate is proven in vivo, the totality of Pla’s substrate synthesized by the host is not necessarily cleaved and the proof of cleavage does not preclude a role in pathogenesis [65,66].

#### 2.1.2. Immune-Related Targets

Several investigations indicated that the proteolytic activity of Pla interferes directly with the host immune response in addition to fibrinolytic and coagulation pathways. Convincing evidence came from peptide array-based approach and comparative proteomic analysis using cell-free bronchoalveolar lavage fluid (BALF) samples aiming at discovering putative host substrates of Pla [66,67]. These studies unveiled a little over a dozen Pla targets, including several previously known targets. Interestingly, the majority of the putative substrates identified in this study have a role in immune system processes or proteolysis, suggesting that Pla is dedicated to inducing overwhelming proteolysis but also to affect the immune response. In this regard, it is important to remember that the fibrinolytic/coagulation pathways are intimately associated with the host immune response, suggesting Pla may disrupt the host defense from different sides. Among the proteins related to the immune response and confirmed to be substrates for Pla are the previously identified complement components 3 and 4b, the Peroxiredoxin-6 (Prdx6), and the Fas ligand (FasL) [6,66,67]. Although C3 and C4b play a central role in the activation of the complement cascade, their degradation by Pla neither confers any advantage regarding serum resistance nor seems to interfere with the opsonophagocytosis associated with the complement cascade [6,60]. PrdX6 is a glutathione peroxidase carrying phospholipase A2 activity that plays an antioxidant role in the lungs, notably under various stress conditions [68,69,70,71]. Pla cleaves this protein at sites Lis_173_/Arg_174_, Lys_201_/Leu_202_, and at the undefined site located in the C-terminal region, leading to the disruption of both enzymatic activities of PrdX6 [66]. FasL is a type-II transmembrane protein that induces apoptosis upon interaction with the Fas receptor. This induced cell death program, which implies the caspase-8 followed by the caspase-3/7 pathway, plays an important role in the regulation of the immune system, avoiding excessive immune response and protecting against various infections [72,73,74]. Hence, the proteolysis at multiple sites located within the extracellular domain of FasL by Pla annihilates the capacity of FasL to induce apoptosis. In addition to cleaving proteins, Pla was proposed to cleave antimicrobial peptides. This idea originated from an experiment aiming at understanding how *Y. pestis* can resist the action of the antimicrobial peptides contained in the rat BALF [75]. After showing that antimicrobial peptides containing in the BALF kill *Y. pestis*, the proteolytic activity of Pla was assumed to confer some protection against antimicrobial peptides displaying a putative cleavage site for Pla (i.e., two consecutive basic amino acid residues), such as LL37 [75,76]. However, the role of Pla was unveiled only when *Y. pestis* was unable to produce the pseudocapsule F1, and formal proof of peptide degradation was not given [75]. In other words, it is possible that *Y. pestis* proteins processed by Pla could sensitize the bacillus to antimicrobial peptides. Indeed, several chromosomally encoded proteins were found to be cleaved by Pla [77]. To date, in addition to Pla, 4 of these proteins were identified. They are named KatY, YapA, YapE, and YpaG. 

#### 2.1.3. Ypestis Proteins Processed by Pla 

YapA, YapE, and YapG belong to the type V autotransporter family [78,79]. The proteins of this family possess all the domains needed for their own translocation across the bacterial membranes [80]. Notably, a central domain harboring the function of the mature protein (referred to as the passenger domain) is located between the *N*-terminal signal peptide (allowing the secretion through the inner membrane via the Sec system) and the C-terminal domain that mediates translocation of the passenger domain across the outer membrane. Once translocated, the three Yap autotranspoters are cleaved by Pla [78,79,81]. Notably, Pla processes YapG at multiple sites (Lys_512_, Lys_548_/Lys_549_, Lys_594_/Lys_595_, Lys_558_, and Lys_604_) and YapE at two sites (Lys_232_ and Lys_338_ but preferentially at Lys_232_) (see Table 1). 

Consequently, different portions of the passenger domain of YapG and YapE are released into the environment, but a significant portion of the YapE passenger domain remains associated with the bacteria. Interestingly, this portion is required for autoaggregation and adherence to eukaryotic cells, which presumably explains why YapE is important for bubonic plague [78,79,81,84]. Therefore, processing of the surface Y. pestis proteins by Pla can unveil functions important for pathogenesis. In addition to the ability to degrade chromosomally encoded proteins, Pla was also reported to cleave in vitro a set of plasmid-encoded proteins, the Yersinia outer membrane proteins (Yops) [14,26,38,47,86,87]. However, the degradation of Yops does not occur in a condition mimicking the phagolysosome [77]. The Yops are exotoxins secreted by a type three secretion system from the bacterial cytoplasm into the host (mostly into phagocytes) to disrupt the mammalian immune system, by inhibiting phagocytosis and production of pro-inflammatory cytokines and inducing apoptosis and section anti-inflammatory cytokines [88]. 

### 2.2. Overall Interactibility of Pla 

Pla is a multifunctional protein, which in addition to its role in the cleavage (activation) of plasminogen to generate plasmin [5,89] is also important for suppression and evasion of the innate immune responses [6,90]. Furthermore, Pla can control the development of primary pneumonic plague [91,92], can degrade several potentially important mammalian proteins [93,94], and even act as both an adhesin and invasin, mediating tight binding to fibronectin and promoting the uptake of the bacteria by non-phagocytic cells [94,95]. Often, protein multifunctionality is determined by the presence of IDPRs, which are capable of promiscuous binding to multiple unrelated partners, which enables them to function in regulation, signaling, and control, where they are commonly engaged in one-to-many and many-to-one interactions [96,97,98,99,100,101,102,103,104,105,106,107]. 

To further illustrate this idea, Figure 4 represents the results of the in silico analysis of the interactability of Pla from *Y. pestis*. The resulting Pla-centered network contains 49 nodes (proteins) connected by 247 edges (protein-protein interactions). In this network, the average node degree is 10.1 (in other words, on average, each protein in this network is involved in an interaction with at least 10 other members of the network). Furthermore, the average local clustering coefficient (which defines how close its neighbors are to being a complete clique) is 0.792; a coefficient of 1 means that every neighbor connected to a given node Ni is also connected to every other node within the neighborhood, and a zero value indicates an absence of connection between the nodes. The number of interactions cited above is unexpectedly high since the number of interactions among proteins in a similar size set of proteins randomly selected from the human proteome is equal to 102. In other words, the Pla-centered PPPI network has significantly more interactions than expected, being characterized by a PPI enrichment *p*-value of <10^−16^. Therefore, Pla could be considered as a hub protein capable to degrade or interact with many *Y. pestis* proteins.

### 2.3. Structure and Catalytic Mechanisms 

In contrast to OmpT from *E. coli*, Pla cleaves α2-antiplasmin, and has a stronger ability to activate plasminogen [28]. The tertiary structure of Pla and OmpT is represented by ten transmembrane β-strands and five loops located on the cell surface (L1–L5) (see Figure 1). This preservation of the structure makes it possible to generate Pla-OmpT hybrids to identify the key elements that explain the difference in the enzymatic activities of these two omptins. Analysis of hybrid Pla-OmpT proteins constructed by exchanging surface loops between Pla and OmpT showed that L3 and L4 loops are responsible for the substrate specificity of Pla with respect to α2-antiplasmin [28]. Consistently, the crystallographic analysis revealed that loops L3 and L4 form the entrance to the active site [18]. Replacement of 25 surface-located amino acid residues made it possible to establish that residues His_101_, His_208_, Asp_84_, Asp_86_, Asp_206_, and Ser_99_ are necessary for the proteolytic activity of Pla [28]. Furthermore, unidentified residues in the L3 loop and the residue Arg_211_ located in the L4 loop and pointed inward to the active site play a role in the cleavage of α2-antiplasmin and activation of plasminogen [18]. Interestingly, the conversion of the OmpT protein to a protease similar to Pla in its enzymatic activity was achieved by removing residues Asp_214_ and Pro_215_, replacing Lys_217_ with Arg_217_ in the L4 OmpT loop, and replacing the entire L3 loop with the analogous one from Pla [28]. Surprisingly, only 2 out of the known 7 residues required for plasminogen activation are needed for activation of the host plasminogen uPA [61]. Furthermore, these mutations did not abrogate the cleavage activity against the inactive form of uPa [61]. Therefore, Pla could have different active sites, which could explain why Pla is such a broad-spectrum protease. Figure 5 shows the zoomed-in active site of Pla from *Y. pestis* and illustrates peculiarities of the local environment of four catalytic residues of this protein, Asp_84_, Asp_86_, Asp_206_, and His_208_.

### 2.4. Pla, an Adhesin with Multiple Binding Substrates 

Although Pla was initially identified as a protease, a study that screened a cosmid library for *E. coli* clones containing *Y. pestis* genes that confers the ability to bind type IV collagen, identified Pla as an adhesin [15]. Its adhesiveness has been documented with human, but not with murine collagen, however [15,109]. 

Further investigations that assessed the adhesiveness of purified Pla or *Y. pestis* or *E. coli* expressing *pla* to different substrates revealed that the protein binds weakly to fibronectin, glycosylated type I, type IV, and type V collagens and strongly to laminin (one of the major extracellular matrix component) [109,110]. Consequently, Pla has been shown to be important for binding to different types of epithelial and endothelial cells, as well as diverse types of macrophages [38,50,95,111,112]. Interestingly, binding to extracellular matrix components and cells is independent of the proteolytic activity but subject to environmental conditions (pH and temperature) [38,111]. Pla binds to laminin more efficiently if *Y. pestis* is cultured at 37 °C rather than ≤28 °C, especially if the bacteria are incubated at pH 7 rather than pH 6 [38,113]. Thanks to the phage display and alanine-scanning mutagenesis, the motif Leu_65_-Thr_66_-Leu67 as well as the residues Gly_178_ and Leu_179_ were found to play a role in the interaction with laminin [113]. However, the presence of these residues in the region of Pla embedded in the membrane suggests that they are not directly involved in binding. Lastly, Pla may also be an invasin (i.e., a protein that promotes the entry into host cells) because its presence, under specific conditions, promotes invasion of HeLa cells and human type I pneumocytes [95]. Pla also promotes invasion of alveolar macrophages via the C-type lectin receptor, DEC-205 (CD205), which has an important role in the antigen-presenting process present on many APCs in both human and mice [114]. Lastly, like the adhesiveness property, the invasiveness capacity of Pla is independent of its proteolytic activity, since a substitution mutation at the residues Ser_99_ or Asp_206_ that abrogates the Pla’s proteolytic activity does not impact the capacity of *E. coli* expressing Pla to invade human endothelial cells [111]. Hence, a link between Pla-mediated adhesion and invasion likely exists.

## 3. The Biological Role of Plasminogen Activator in Pathogenesis

### 3.1. The Proven or Unproven Role of Pla

Thanks to its proteolytic and adhesin/invasin activities, Pla is a Swiss army knife of *Y. pestis*’ life cycle. However, *Y. pestis* uses Pla to infect the mammals, but not the fleas [6,115]. Furthermore, our knowledge of the Pla-mediated molecular mechanism used by *Y. pestis* to produce infection remains limited, notably for bubonic plague. We do not know whether the Pla-invasin mediated activity is important for the plague. In contrast, *Y. pestis* has been demonstrated to use the Pla-mediated adhesion and proteolytic activities to establish an infection in mammals. However, the adhesin activity was studied only in pneumonic plague [116]. It is also worth noting that not all of Pla’s targets identified in vitro seem to be processed in vivo. For instance, the α2-antiplasmin does not appear to be a substrate during pneumonic plague [65]. In fact, PAI-1 and Fas ligand are the two only substrates for Pla confirmed in vivo [62,67]. Lastly, indirect evidence suggests that Pla degrades Prx6 and YapE in vivo. Notably, Pla may cleave Prdx6 during the pneumonic plague, because the amount of extracellular Prdx6 is lower in the lungs of mice infected with wild-type strain than in animals infected with a Δ*pla* mutant [66]. Whether this cleavage occurs really or not, it does not play any obvious role in the pathogenesis of pneumonic plague. As for YapE, it might be cleaved at least in the skin and/or in the course of lymph node colonization. In fact, a Δ*yapE* mutant displays similar dissemination defect as a Δ*pla* mutant after subcutaneous inoculation of bacteria [78].

### 3.2. The Role of Pla during Bubonic Plague

Initially, it was suggested that *Y. pestis* uses the coagulase and fibrinolytic activity of Pla to produce a transmissible infection in flea and to initiate the host colonization at the fleabite site, respectively [6,117]. In particular, *Y. pestis* would have used the coagulase activity (Pla is active at ≤28 °C) to produce a fibrin clot that obstructs the digestive tract of the flea. This process increases the likelihood of *Y. pestis* transmission by fleas [118]. Once transmitted to a new mammalian host, *Y. pestis* would have used the fibrinolytic activity of Pla (active at ≥37 °C) to prevent the formation of a blood clot produced at the insect’s suction point; this clot normally traps bacteria and prevent them from spreading. Hence, *Y. pestis* would use Pla to degrade fibrin clot, extracellular matrix, and basement membrane (i.e., the main obstacle to the spread of microorganisms) to escape from the skin and reach the drainage lymph node, where its active replication leads to the production of an enlarged painful lymph node, the bubo (the hallmark of bubonic plague). In such a model, *Y. pestis* cannot produce a transmissible infection in flea or a flea-borne infection in the absence of Pla. However, experiments performed in the *Xenopsylla cheopis* rat flea rejected the potential Pla’s role in this insect [36,115]. Conversely, the role of Pla in mammals has been confirmed on multiple occasions [6,60,90]. Its deletion results in a one million-fold reduction in the bacterial virulence in mice inoculated by a subcutaneous route. Histological analysis indicates that a Δ*pla* mutant does not reach deep tissue when injected at a dose equivalent to that of regurgitated by fleas [6,60,90,119]. Virulence studies utilizing plasminogen-deficient mice supported the idea that *Y. pestis* uses Pla to degrade the plasminogen, and so the uncontrolled production of active plasmin, to produce bubonic plague [89]. Indeed, plasminogen-deficient mice are resistant to *Y. pestis*. However, a *pla*-negative *Y. pestis* mutant remains avirulent in plasminogen-deficient mice. Therefore, whether Pla cleaves the plasminogen during the bubonic plague, this cleavage can only partially explain the role of Pla in this form of the disease. The proteolytic activity of Pla against multiple complement regulatory proteins could have been incriminated to explain why a Pla-negative *Y. pestis* remains avirulent in plasminogen deficient mice. However, *Y. pestis* does not use Pla for serum resistance [6]. Furthermore, virulence studies in congenic C5A+ and C5A- mice revealed that Pla-associated virulence is not mediated by interference with the phagocytic chemoattractant C5a [60]. Lastly, it is worth reminding that *Y. pestis* cleaves YapE thanks to Pla, and a Δ*yapE* mutant displays a similar dissemination defect as a Δ*pla* mutant after subcutaneous inoculation of bacteria [78]. Hence, *Y. pestis* may use Pla to cleave its own proteins rather than the host proteins to produce bubonic plague. Overall, it is clear that the molecular mechanisms leading to bubonic plague and involving Pla remain to be elucidated.

Based on the above-cited virulence data, it was believed for many years that flea-borne plague is impossible if the bacillus lacks Pla. However, this view is somewhat challenged by the presence of natural isolates that lack Pla (0.PE2 or bv. Caucasica) and are fully virulent [120,121]. In addition, mice challenged with fleas infected with a mutant Δ*pla* die of primary septicemic plague because fleas seem to regurgitate bacteria in the extravascular part of the dermis in 70–90% of the cases and the intravascular part of the dermis in 10–30% [90,122,123]. In a natural context of infection (i.e., fleabite), a Δ*pla* mutant can sometimes spread to regional lymph nodes, as others have observed after subcutaneous or intradermal inoculation. However, the infection does not progress to severe lymphadenitis and bubonic plague. Pla, therefore, increases the incidence of plague by allowing the production of the bubonic form of the disease. In contrast to the model suggesting an essential role of Pla for the spread of *Y. pestis* from the skin, one study suggested that Pla is not needed for infiltration and destruction of the draining lymph node [124]. Indeed, the Δ*pla* mutant colonized and destroyed this organ in a similar manner to the wild strain when it was injected at a relatively high dose (5000 CFU) into the pinna of the mouse ear. In fact, Pla was proposed to be a key genetic component necessary to survive and multiply in the draining lymph node. This idea was partly based on immunostaining which revealed an abnormal bacterial morphology. However, the use of an anti-*Y. pestis* antiserum and not a specific antibody precludes to definitively conclude about this idea. Furthermore, one may question why the mutant will be destroyed in the lymph node but not in the skin and deep tissues. In this context, it is worth noting that discrete clumps of Δ*pla* bacteria were present only in the marginal sinus of the lymph node of moribund mice infected by fleabite [90]. Furthermore, these bacterial clumps were in close contact with fibrin deposit and inflammatory cells, suggesting that *Y. pestis* uses Pla to escape from fibrin clots in the lymph node after fleabite. Therefore, the role of Pla in a natural vs. artificial context of infection could be different. Alternatively, the low number of lymph nodes analyzed in mice that developed fatal plague after fleabite may have revealed just one of Pla’s multiple roles in the pathogenesis.

### 3.3. The Role of Pla during Primary Pneumonic Plague 

Although Pla is essential for producing bubonic plague after a fleabite, it is not crucial for producing a fatal infection after inhalation of *Y. pestis* [92]. Its loss slightly delays the mortality of mice inoculated by the intranasal route. However, mice that die after such an inoculation succumb to septicemia and not to pneumonia. In other words, Pla is dispensable for dissemination into the bloodstream from the lungs but is essential for the outgrowth of *Y. pestis* in the lungs, which leads to edema, severe tissue damage, and immune cell influx, i.e., primary pneumonic plague. Hence, Pla appears essential for aerosol transmission of the disease, i.e., human-to-human transmission. A series of elegant investigations suggest that Pla plays a role during all stages leading to pneumonic plague [44,62,67,112,116,125,126,127]. During the early stages, *Y. pestis* appears to use the adhesin and the proteolytic activity of Pla to temporize the immune response, thus to establish the colonization [112,116]. Notably, *Y. pestis* binds to alveolar macrophages thanks to Pla. This Pla-mediated adhesion aids *Y. pestis* to translocate the Yop exotoxins into the cytoplasm of alveolar macrophages, and thus inhibit the production of an inflammatory response and recruitment of PMNs. However, since the inhibition is never complete, some PMNs are still recruited. *Y. pestis* resists these recruited cells via the proteolytic activity of Pla. The Pla’s substrate leading to this resistance is unknown. One may suggest that they are some of the extracellular effectors known to be secreted by PMNs in infected tissues [43,75,76,128]. Regardless of the exact mechanism of how *Y. pestis* resists the low influx of PMNs in the lungs, this resistance, combined with Pla-mediated adhesion, creates a niche characterized by relatively low inflammation, where *Y. pestis* rapidly proliferates.

As a rule, the initial suppression of the host’s immune response by *Y. pestis* is never complete; so eventually the active replication of bacteria is recognized by the immune system. Hence, the inflammation is amplified and the host initiates a strong immune response. During this high inflammation stage, *Y. pestis* transcribed more actively *pla* (via CRP) in response to the reduction of glucose concentration in the lungs [48] and uses Pla to cleave FasL and PAI-I (whose level increased in response to the presence of *Y. pestis* in the lungs) [62,67]. The cleavage of FasL disrupts the Fas-FasL signaling, which normally induces cellular apoptosis dependent on the caspase-3/7-dependent pathway to mount an efficient innate immune response. Hence, the cleavage of FasL alters cytokine production, immune cell recruitment, and lung permeability, which altogether would allow a high proinflammatory response and tissue damage leading to pneumonia. In contrast to the inactivation of FasL, the inactivation of PAI-I by Pla does not seem to play a major role in pneumonic plague [62,67]. Indeed, the kinetics of death in PAI-1 KO and WT mice infected by the intranasal route with the Δ*pla* mutant was similar; i.e., the loss of PAI-1 did not restore a wild-type phenotype of Δ*pla* mutant. Furthermore, wild-type *Y. pestis* grows similarly in the lungs of WT and PAI-1 KO mice and causes less severe lung damage in PAI-1 KO than WT mice, presumably due to lower ability to recruit PMNs in PAI-1 KO mice. This later data was unexpected as the inactivation of PAI-1 by Pla is thought to enhance plasmin generation, thus facilitating bacterial outgrowth and dissemination. Paradoxically with the less severe lung injury and reduction of immune cell recruitment observed in PAI-1 KO mice, the cleavage of PAI-1 by Pla has been associated with increased production of proinflammatory cytokines. This suggests that the cleavage of PAI-1 by Pla may play a role during lung colonization and/or pneumonic plague production that is hidden by a major process, such as that caused by the Pla-mediated cleavage of FasL.

## 4. The Evolution Aspect

Comparative genomic analysis provided deep insight into the emergence of *Y. pestis* and its microevolution [129,130]. The different strains of *Y. pestis* can be grouped into two distinct subspecies. One subspecies, named *microti*, includes the oldest among currently living strains, which can be further grouped into distinct biovars: Altaica (0.PE4), Qinghaiensis (0.PE4ab), Xilingolensis (0.PE4cd), Talassica (0.PE4), Hissarica (0.PE4), and Ulegeica (0.PE5) (Figure 6). These strains circulate in populations of various species of voles (*Microtus* spp.) and are highly virulent for their main hosts and laboratory mice. However, as a rule, such strains are avirulent for guinea pigs and do not cause epidemic outbreaks in humans [131]. The second *Y. pestis* subspecies, named *pestis*, is composed of strains grouped into four distinct biovars: Antiqua (0.ANT-4.ANT), Medievalis (2.MED), Orientalis (1.ORI), and Intermedium (1.IN). These strains can infect a large variety of mammalian species and some of them were associated with the three plague pandemics [2]. 

Interestingly, some *microti* strains, namely those belonging to the biovar Caucasica (0.PE2 cluster), lack *pla* (Figure 6) [120,135,136]. Furthermore, Haiko et al. showed that some *microti* stains, representatives of biovars Angola and Xilingolensis, synthesize a Pla displaying isoleucine at position 259 instead of the threonine present in Pla from all “modern” isolates [35,137]. The absence of *pla* and the presence of the I259 isoform of Pla was further confirmed in a survey of 100 *microti* strains belonging to the 0.PE cluster [131]. This later study confirmed that all stains of the biovar Caucasica (0.PE2) lack *pla* whereas all the stains of the biovars Altaica (0.PE4), Qinghaiensis (0.PE4ab), Xilingolensis (0.PE4cd), Talassica (0.PE4), Hissarica (0.PE4), and Ulegeica (0.PE5) express the I259 isoform. The Bronze-Age (0.PRE1, 0.PRE2) SNP-based *Y. pestis* genotypes [138], as well as the Neolithic-lineage strain [133], were also found to exhibit the ancestral *pla* allele. Hence, comparative genomic studies suggest that I259 Pla is an ancient isoform, and this later form gave rise to the T259 isoform. Interestingly, the I259T substitution is associated with a better ability to activate plasminogen, as well as a decrease in the ability to inactivate α2-antiplasmin [35]. 

Altogether, in silico analysis and our knowledge regarding the host spectrum of the different *Y. pestis* subspecies provided a strong argument in favor of the decisive importance of the acquisition of Pla and/or the I259T substitution for an increased incidence of plague and/or the transition of the plague microbe from a “narrow” to a "broad" mammalian host spectrum. Consistent with this idea, flea-borne transmission of *Y. pestis* leads to the production of fatal primary septicemic plague (septicemia without bubo) and fatal bubonic plague (septicemia with bubo) and only this later form was *pla*-dependent [90,122,123]. Therefore, the acquisition of *pla* appears to increase the incidence of flea-borne plague by allowing the bacteria to overwhelm the lymph node draining the fleabite site. Although important for the production of fatal bubonic plague, the acquisition of Pla was not essential for the infection by the intranasal route. However, its acquisition would have resulted in the production of pneumonia characteristic of pneumonic plague, i.e., production of a rapidly fatal disease that can be transmitted by aerosol [139]. Consistent with the idea that the I259T substitution provided a selective advantage, virulence testing in mice infected with variants of the Pestoides F (Pla^−^, *microti* lineage) or CO92 (T259 Pla, *pestis* lineage) strains, differing in the presence or absence of one of the Pla isoforms, concluded that the presence of the T259 Pla isoform significantly enhances the invasive ability of *Y. pestis* during bubonic plague [139]. However, a different conclusion was drawn from an investigation using Pestoides F analog. The expression of the T259 Pla isoform in this *microti* strain does not change its invasiveness or its ability to kill mice and guinea pigs [140]. However, both studies agreed that the I259T substitution reduces the time to death in mice, indicating that substitution accelerates the transmission rate of the bacillus at least in some host species. However, the hypothesized role of Pla and its various forms in the emergence of plague are somewhat weakened by the fact that the deletion of pPst in (at least several) modern subsp. *pestis* strains did not impact the virulence in mice and guinea pig infected by cutaneous and by aerosol route [136,141]. Furthermore, DNA of the oldest *Y. pestis* lineages containing the ancient allele of the *pla* gene is found in the bones of people who died in the Bronze and Neolithic Ages [132].

It should be noted that Pla is not necessarily essential for all strains of *Y. pestis*. Indeed, the 0.PE2 natural isolates do not have the pPst (harboring Pla). Furthermore, a number of strains cured in the laboratory of the pPst plasmid retain virulence at the level of wild-type strains. This dependency of Pla according to the strain considered could be due to the presence of another pathogenicity factor encoded by additional plasmids or inserted in the chromosome. Alternatively, a mutation could compensate for the absence of Pla. However, 0.PE2 natural isolates, as well as strain 231 (0.ANT3), lack additional cryptic plasmids, and the amino acid sequences of the *Y. pestis* Pla-like protein (YcoA) is 100% identical to that of strains for which Pla-negative derivatives practically completely lost their virulence. It, therefore, remains to be elucidated why some strains of *Y. pestis* depend on Pla and others do not.

## 5. Pla: Not a Vaccine Antigen, Maybe an Antibacterial Target, but a Diagnosis Tool

One may suggest that Pla antigen may be relevant to plague resistance because convalescent animals or patient victims of plague or vaccinated with a vaccine strain developed antibody raised against this outer membrane protein [142,143,144,145]. However, anti-Pla titers were relatively low and early antibiotic treatment as well as the antibiotic used for treatment reduced the percent of animals mounting an anti-Pla response, indicating that Pla may not be a protective antigen [143]. Consistently, the immunization with Pla (purified from recombinant *E. coli*) is barely immunogenic and unable to confer protection against the different forms of plague in different animal models [143,144,146]. Furthermore, a Pla-based DNA vaccine was also unable to confer protection against the disease, even after optimization which increases the DNA vaccine immunogenicity [146]. Yet, DNA can be highly effective in the induction of cell and humoral mediated responses. Thus, Pla does not appear to be a protective antigen against plague. This conclusion does not mean that interest in *pla,* or rather its absence to produce an anti-plague vaccine, is meaningless. Strains attenuated in virulence due to the absence of *pla* replicate in the skin after subcutaneous inoculation. Furthermore, they show a transient replication at peripheral sites when injected at high doses. Thus, a Pla mutant might stimulate a vigorous immune response. This information added with the fact that Pla is not an effective immunogen and is not a protective antigen in mice suggest that the absence of *pla* should not decrease the efficacy of a live plague vaccine while increasing its safety. Consistent with this idea, different vaccine strains were engineered [147,148,149,150]. One of them, lacking the Braun lipoprotein, the lipid A biosynthesis myristoyltransferase and Pla (i.e., a Δ*lpp* Δ*msbB* Δ*pla* strain), confers long-term humoral and cell-mediated immune responses and full protection against pneumonic plague in rats [149].

Considering its critical role in the pathogenesis of plague, Pla has been proposed to be an interesting protein to target for treatment. Notably, it has been suggested that the inhibition of *pla* would expand the window during which treatment of primary pneumonic plague could be successfully administered to treat the disease because the deletion of *pla* increases the time to death [92]. As a proof of concept, the repression of *pla* transcription during the early stage of primary pneumonic plague in mice was shown to extend significantly the time to death. As Pla plays a dual critical role in establishing lung infection during the early stages of disease [116], it shall now be interesting to test whether inhibition of Pla at later time points (i.e., after prophylaxis time or just at the time of curative treatment) can really improve survival rate. 

Although a therapeutic approach based on Pla has yet to be demonstrated, the use of Pla and its product as a means of identifying the presence of the bacterium has proven to be effective. The *pla* gene was proposed to be an interesting target to detect the presence of *Y. pestis*. Indeed, this gene is important for infection, present in high copy number (as high as 186 per bacterium) and it was believed for a long time that it is absent from closely related *Yersinia* species [6,151]. Hence, PCR assay was proposed to be a more rapid and sensitive alternative to the laboratory diagnosis of plague based on pathogen isolation and identification. PCR assay was also more rapid and sensitive than a previous approach based on DNA probe hybridization [152]. Multiple PCR assays were developed to detect the presence of single *pla* or in combination with other genes (i.e., simplex or multiplex PCR). The initial PCR-based approach implied the detection of amplified fragment after migration of the amplicon in an agarose gel followed by gel staining [153,154,155,156]. Although useful, the PCR assay suffers from its reduced ability to perform high throughput testing and is associated with the potential for contamination between samples handled. In other words, this diagnostic method is particularly not well suited to epidemics. Real-time PCR alleviated this issue. Based on fluorescence emission, this later technology allows simultaneous amplification and detection without the need for an agarose gel and reduces the risk of contamination [157,158,159]. “Simple” and real-time PCR assays had been and are used around the world to detect the presence of *Y. pestis* in the environment, several rodent species, fleas, primate, and human samples [153,156,159,160,161,162,163,164,165,166,167,168,169,170]. Fascinatingly, PCR targeting *pla* was used to diagnose septicemia caused by *Y. pestis* in humans that died of plague during ancient pandemics, and so unveiled the history of plague [171,172,173]. 

PCR is a great tool but the risk of false-positive is inherent to this technology. This is why the branched DNA technology (bDNA), a non-PCR-based detection of sequence, was proposed as an interesting alternative [174,175,176]. This technic implies a cascade of target-specific probes that hybridize with other probes, resulting in the amplification of a signal that can be quantified. The use of bDNA targeting Pla was shown to be very sensitive [177]. Yet, this approach was not developed further.

In addition to the *pla* gene, the detection of the enzyme itself was also proposed to identify *Y. pestis*. Thanks to a library of monoclonal antibodies (MAbs) directed against different Pla epitopes, specific antibodies raised against the proteins from *Y. pestis* was generated [178,179,180]. The selected MAbs detect successfully natural *Y. pestis* isolates from the sputum of pneumonic plague patients and the liver and spleen of rodents [179,180]. Furthermore, the selected MAbs are part of a rapid immunoassay test, usable under field conditions [180]. However, the assay suffers from its limit of detection (10^6^ bacteria).

A study of the immune response unveiled that Pla is a predominant antigen in laboratory and wild animals and in humans that survived plague or which were vaccinated with live plague vaccine [142,143,145,181]. These data suggest that a serodiagnosis based on Pla could be of interest after epidemics. Consistently, a report mentioned that some of several captured animals are serologically positive for *Y. pestis* whereas the fleas associated with serologically positive and negative captured animals are free of *Y. pestis* [182]. However, a comparison of immune responses in human volunteers vaccinated or not with live plague vaccine showed that response to Pla can be aspecific. On the other hand, mapping of the cross-reactive regions of Pla identified one specific peptide (PNAKVFAEFTYSKY) that reacts with 50% of sera from vaccinated individuals [183]. Furthermore, unlike the unvaccinated, the vaccinated individuals produced IgA, IgE, IgG3 against Pla, and their PBMCs release IL-17A upon stimulation with Pla antigen. Thus, a Pla-based serodiagnosis could be developed but its development does not seem trivial to put in place.

It is undeniable that *pla* and its product may be of interest for detecting the past or present presence of *Y. pestis* in the environment, in sick or convalescent animals or humans. However, basing a diagnosis, whatever its nature, solely on the Pla seems foolhardy. Indeed, as we stressed above, Pla is absent in some *Y. pestis* strains [120,135,136]. Furthermore, Pla is not essential to produce a lethal infection by aerosol and flea bites [44,45]. Worse, Pla is not specific to *Y. pestis*. Indeed, positive PCR and evening sequencing revealed the presence of Pla in European rats and archaeological samples [184,185]. Furthermore, this gene occurs in some *Citrobacter koseri* and *E. coli* strains [185,186]. In this context, it seems essential that diagnostic tests should involve other targets in addition to the *pla* and its product. Hence, the multiplex PCR assays already set up make sense [156,159,162,169,187,188].

## 6. Concluding Remark

In conclusion, a considerable number of investigations on *pla* and its product have provided a nice example of how gene acquisition, gene loss, and point mutation shaped a bacterial genome for the emergence of a pathogen such as *Y. pestis*. Overall, they have also provided a better understanding of the role of Pla in the physiopathology and the molecular mechanisms leading to bubonic and pneumonic plague. However, we still need to perform further investigations in vivo to understand the exact role of the “ancient and modern” Pla proteins (as a protease and adhesin/invasin and its multiple substrates) during the different stages and forms of plague. This is even truer for the progression of bubonic plague in a natural context of infection as flea saliva and Pla both interfere with the coagulation pathway. However, a better understanding of the role of Pla in pathogenesis could arise only if the different investigators compare and apply the same standard operating procedures; to avoid discrepancy conclusions in the literature. Besides a better understanding of plague and *Y. pestis*, a tremendous amount of work also highlighted *pla* and its product as a target of medical importance, notably for diagnosis and putative treatment. However, we have to take this interesting highlight with caution in light of the genetic diversity of the different *Y. pestis* strains and the real medical world; i.e., with regards to our available and efficient antibiotics arsenal for prophylaxis vs. curative treatment, and palliative care that is used by the intensive care unit.

## Figures and Tables

**Figure 1 biomolecules-10-01554-f001:**
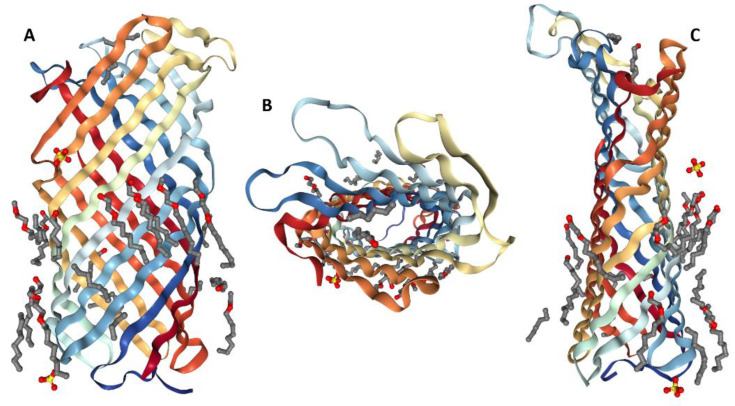
3D structure of the plasminogen activator Pla from *Yersinia pestis* determined by X-ray crystallography (PDB ID 2 × 55) [30]. Two side projections (plots **A**,**C**) and a top view from the extracellular side (plot **B**) of Pla are shown together with the set of C_8_E_4_ detergent molecules.

**Figure 2 biomolecules-10-01554-f002:**
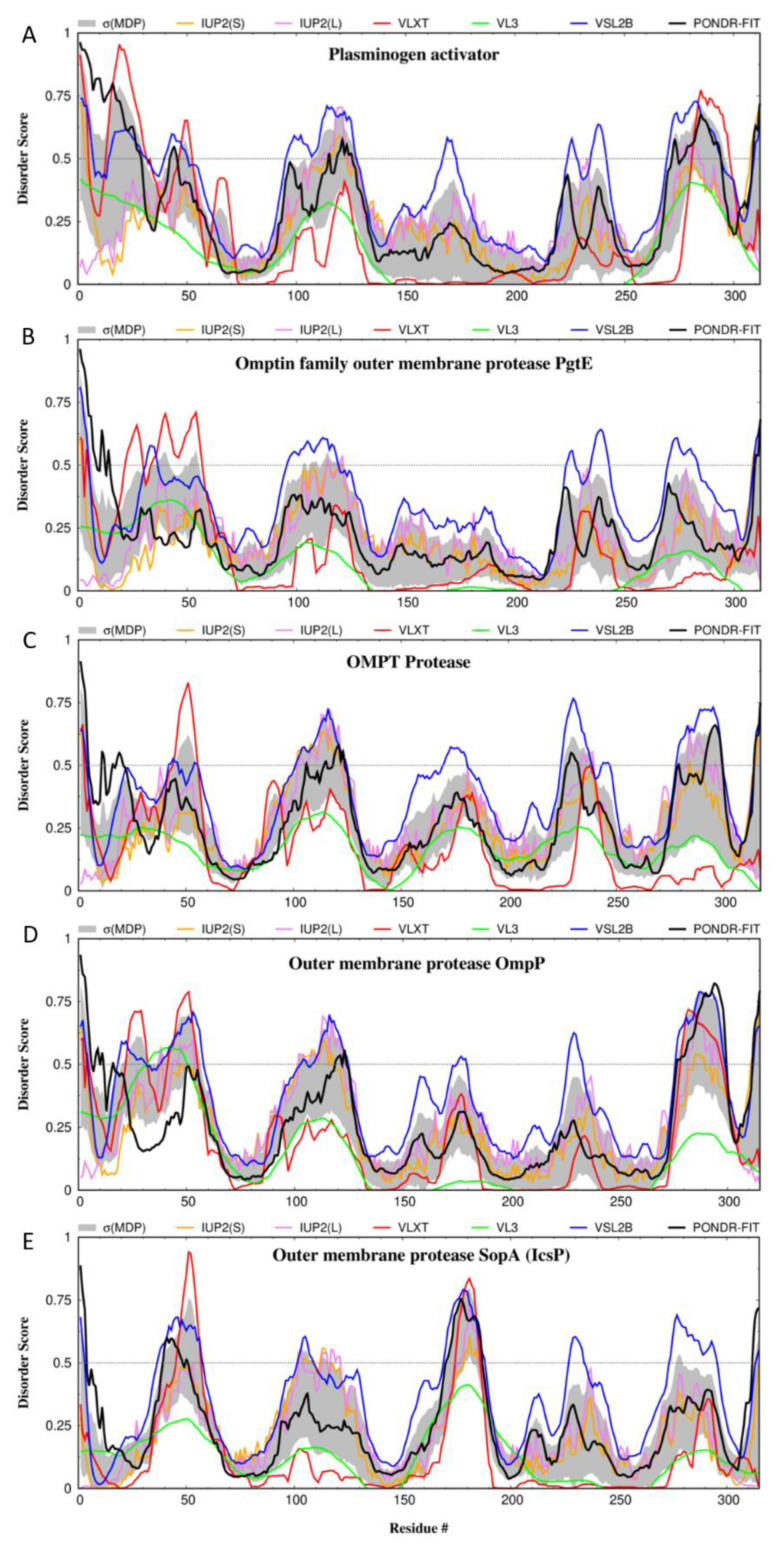
Intrinsic disorder profiles generated for the plasminogen activator Pla from *Yersinia pestis* (**A**), Omptin family outer membrane protease PgtE (UniProt ID: A0A0D6FBC6) from *Salmonella enterica* (**B**), Outer membrane protease OmpT (UniProt ID: P09169) from *E. coli* (**C**), Outer membrane protease OmpP from (UniProt ID: P34210) *E. coli* (**D**), and Outer membrane protease SopA/IcsP (UniProt ID: O33641) from *Shigella flexneri* (**E**) by DiSpi web-crawler. Outputs of different commonly used disorder predictors, PONDR^®^ VLXT, PONDR^®^ VL3, PONDR^®^ VLS2B, PONDR^®^ FIT, IUPred2 (Short), and IUPred2 (Long) are shown by red, green, blue, black, orange, and pink colors, respectively. Gray shaded area represents errors evaluated for mean disorder profile (MDP) calculated by averaging profiles of individual predictors.

**Figure 3 biomolecules-10-01554-f003:**
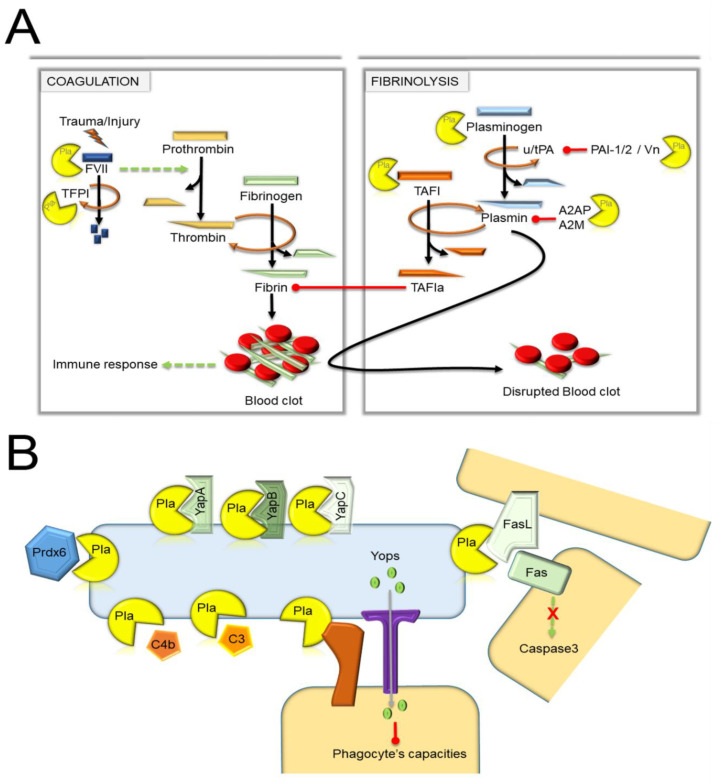
The different substrates of Pla. (**A**) Hemostasis cascade. Trauma or an injury activates the coagulation cascade, which leads to the production of a blood clot. The fibrinolysis inhibits the coagulation cascade and disrupts the blood clot. Dashed green and brown circular arrowheads indicate respectively a multiple steps process and a single step leading to protein cleavage. Black arrowheads show the different products of the cleaved protein. Red lines with ball indicate inhibition. Pla (pacman) acts as a protease. (**B**) The Pla protease cleaves proteins located at the surface of the bacteria (blue cell) and the host’s cell (yellow) or other host’s proteins. Pla also acts as an adhesin (independently of its proteolytic activity), which triggers the secretion of Yops into the host’s cell cytoplasm and inhibits the phagocyte’s ability to kill the bacteria and induce an immune response. Redline with ball indicates inhibition; dashed green arrowhead, a multiple-step process leading to caspase 3 activation; red-cross, inhibition.

**Figure 4 biomolecules-10-01554-f004:**
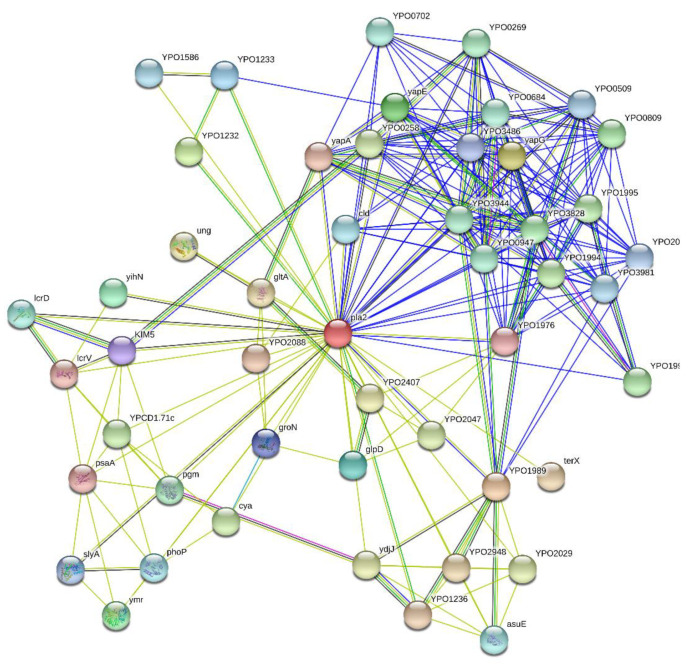
Interactability of Pla with other *Y. pestis* proteins. The Pla-centered protein-protein interaction (PPI) network was generated by the Search Tool for the Retrieval of Interacting Genes; STRING (http://string-db.org/. STRING). This later generates a network of associations based on predicted and experimentally-validated information on the interaction partners of a protein of interest [108]. In the corresponding network, the nodes correspond to proteins, whereas the edges show predicted or known functional associations. Seven types of evidence are used to build the corresponding network, where they are indicated by the different colored lines: a green line represents neighborhood evidence; a red line—the presence of fusion evidence; a purple line—experimental evidence; a blue line—co-occurrence evidence; a light blue line—database evidence; a yellow line—text mining evidence; a black line—co-expression evidence [108].

**Figure 5 biomolecules-10-01554-f005:**
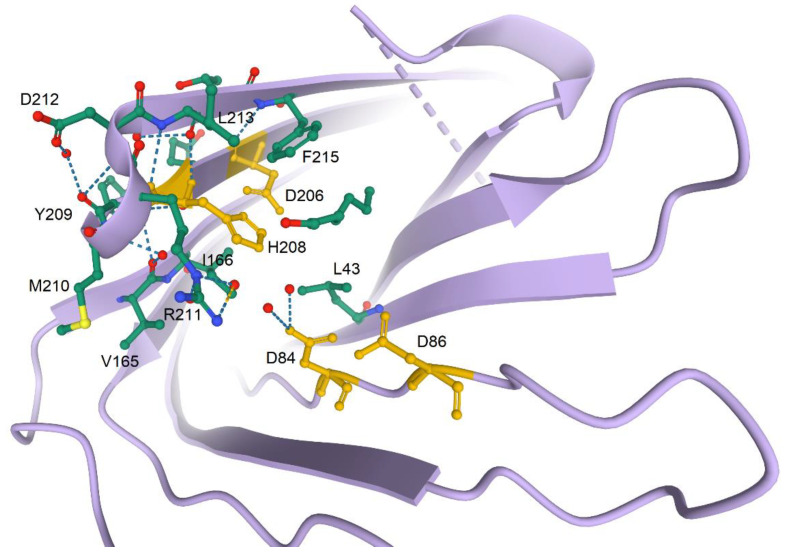
Zoomed in structure of the active site of Pla from *Y. pestis* illustrating some peculiarities of the local environment of four catalytic residues of this protein, Asp_84_, Asp_86_, Asp_206_, and His_208_. X-ray crystal structure of Pla from *Y. pestis* was used here (PDB ID 2X55) [30].

**Figure 6 biomolecules-10-01554-f006:**
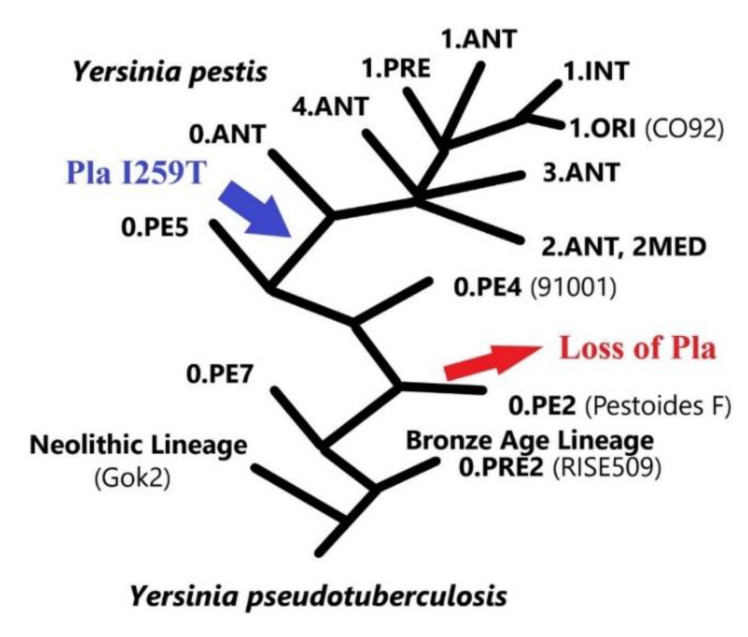
Genomic tree and divergence based on 503 *Y. pestis* Genome Assembly and Annotation reports (https://www.ncbi.nlm.nih.gov/genome/browse/#!/prokaryotes/153/). Both the modern and ancestral phylogenetic groups are indicated. The tree was adapted from Achtman [132] and Rascovan et al. [133]. The relationship between subspecies, biovars, and SNP types is shown by Kislichkina [134].

**Table 1 biomolecules-10-01554-t001:** Pla interactions with different substrates.

Substrates	Biological Function/Process	Hydrolyzable Amino Acids of Substrate	Consequence	Proven or Not Substrate In Vivo	Contribution to Virulence/Pathogenesis	References
Host proteins processed by Pla						
Peroxiredoxin 6 (Prdx6)	Immune system process; ROS metabolic process	Cleaves at sites Lis173/Arg174, Lys201/Leu202, and the undefined site located in the C-terminal region	Disrupt peroxidase and phospholipase A_2_ activities	yes	The cleavage of Prdx6 has a little detectable impact on the progression or outcome of pneumonic plague	[66]
Alpha2-antiplasmin (A2AP)	proteolysis; contributes to control of the pulmonary inflammatory response to infection by reducing neutrophil recruitment and cytokine production	ND	uncontrolled production of active plasmin and resulting clearance of fibrin depositions	no	A2AP is not significantly affected by the Pla protease during pneumonic plague; A2AP participating in immune modulation in the lungs has a limited impact on the course or ultimate outcome of the infection	[65,66,82]
Plasminogen activator inhibitor-1 (PAI-1)	inhibition of activation of plasminogen.	cleaves between residues R346 and M347	prevent inhibition of tPA and uPA	yes	PAI-1 deficiency results in a decreased level of neutrophil influx to the pulmonary compartment during pneumonia. This leads to increased bacterial out-growth, enhanced dissemination, and decreased survival of infected mice	[66,82]
Urokinase plasminogen activator (uPA)	activation of plasminogen	cleaves the single-chain uPA (scuPA) between residues Lys158 and Ile159	cleavage led to the activation of scuPA	no	activates fibrinolysis, cell migration, and tissue remodeling	[61,66]
Complement component C3	cytokine activity; complement activation	ND ^1^	ND	no	cleavage of C3 disrupts chemotaxis of inflammatory cells to foci of infection, leads to disturbances in their phagocytic activity and the inability of the complement system to form the cytolytic end product of the complement system activation, membrane attack complex	[66,82]
Apoptotic molecule Fas ligand (FasL)	immune system process; its binding with its receptor induces apoptosis	cleaves at multiple sites located within the extracellular domain of FasL	ND	no	contribute to the progression of pneumonic plague	[66]
Glutathione S-transferase A3	immune system process; ROS metabolic process	ND	ND	no	ND	[66]
Glutathione peroxidase 3	immune system process; response to toxin	ND	ND	no	ND	[66]
Tubulin polymerization-promoting protein family	cell component; structure	ND	ND	no	ND	[66]
Pigment epithelium-derived factor	protein binding; proteolysis	ND	ND	no	ND	[66]
Alpha-2-HS-glycoprotein	protein binding; immune system process; proteolysis	ND	ND	no	ND	[66]
Glutathione S-transferase Mu 1	immune system process; transferase activity	ND	ND	no	ND	[66]
BPI fold-containing family A member 1 (sPlunc)	immune system response	ND	ND	no	ND	[66]
Carboxypeptidase N subunit 2	immune system process; cytokine-mediated signaling	ND	ND	no	ND	[66]
Sulfated glycoprotein 1	protein binding; lipid transport	ND	ND	no	ND	[66]
BPI fold-containing family b member 1 (Lplunc1)	MAC activation, response to stress	ND	ND	no	ND	[66]
Vinculin	actin binding; cell adhesion	ND	ND	no	ND	[66]
Plasminogen	serine-type peptidase activity; proteolysis	cleavage at a single site between residues Arg561 and Val562 of the proenzyme	activates plasminogen through cleavage this zymogen at a single site	no	activates fibrinolysis	[66,82]
Actin gamma	cell component; structure	ND	ND	no	ND	[66]
Plastin-2	structure; actin binding	ND	ND	no	ND	[66]
Lipoprotein lipase	lipase activity; lipid transport	ND	ND	no	ND	[66]
Phosphoglycerate mutase 1	glycolysis	ND	ND	no	ND	[66]
Complement C4-B	complement activation; signal transduction	ND	ND	no	ND	[66]
Hypoxanthine-guanine phosphoribosyltransferase	monosaccharide metabolic process	ND	ND	no	ND	[66]
Calmodulin	Ca^2+^ binding	ND	ND	no	ND	[66]
Apolipoprotein A-IV	lipid transporter activity; blood circulation	ND	ND	no	ND	[66]
Thrombin-activatable fibrinolysis inhibitor (TAFI)	antifibrinolytic plasma protein	ND	Pla can cleave TAFI near its C-terminus, preventing activation to TAFIa during subsequent incubation with thrombin–thrombomodulin; in addition to the direct inactivation of TAFI by Pla, TAFIa can also be inactivated through proteolysis by plasmin	yes	ND	[82]
Tissue factor pathway inhibitor (TFPI)	TFPI is an anticoagulant protein that reversibly binds to coagulation factor Xa (FXa). This bimolecular TFPI–FXa complex is a potent inhibitor of the procoagulant complex TF:FVIIa (the primary initiator of coagulation in vivo), which acts to block further coagulation at this point in the cascade	Cleavage of TFPI by Pla occurs between residues K249 and G250	cleavage by Pla is predicted to have procoagulant consequences; Pla disrupts the TFPI-mediated inhibition of clot formation	no	TFPI inactivation enhances coagulation	[82]
Cathelicidins	cationic antimicrobial peptides (CAMPs)		CAMPs permeabilize bacterial lipid bilayers, resulting in the lysis of affected cells; Pla inhibit CAMPs chemoattractant properties that recruit neutrophils, monocytes, and T cells in response to infection	no	ND	[82]
α-2-macroglobuline	impede the plasmin activity	ND	ND	no	activates proteolysis	[28]
*Y. pestis* proteins processed by Pla						
Type-III secretion system effectors	inhibit phagocytosis, induce apoptosis of macrophages, destroy actin cytoskeleton and signaling pathway of activation of inflammatory cells, suppress production of cytokines and chemokines	ND	degrades most Yops *in vitro* including the Yops B, C, D, E, F, H, J, and M, but is unable to degrade LcrV	no	it is supposed that Pla coordinates the degradation of extracellular Yops that may otherwise compromise innate immunity evasion	[14,82]
YapA	autotransporter	processes at multiple sites (Lys_512_, Lys_548_/Lys_549_, Lys_594_/Lys_595_, Lys_558_, and Lys_604_)	cleavage at the C terminus released the protein from the cell surface	no	it is supposed that YapA might be an adhesin	[83]
YapG	autotransporter	processes at multiple sites	ND	no	does not contribute to *Y. pestis* virulence in established mouse models of bubonic and pneumonic infection	[81]
YapE	autotransporter	processes at two sites (Lys_232_ and Lys_338_ but preferentially at Lys_232_)	cleavage is required to proteolytical activation of the protein	no	contributes to disease in the mouse model of bubonic plague by mediating bacterial aggregation and adherence to eukaryotic cells	[84]
KatY	catalase-peroxidase	Cleavage of α-KatY (78.8 kDa) by Pla resulted in its smaller forms, β-KatY (∼50 kDa), γ-KatY (∼36 kDa) and δ-KatY (∼34 kDa)	ND	no	ND	[85]

^1^ “ND”, no data.

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
