# Peer review of "Yersinia pestis Plasminogen Activator"

_biomolecules, 2020, doi:10.3390/biom10111554_

Round 1

Reviewer 1 Report

This is a long-awaited and comprehensive review on Yersinia pestis plasminogen activator, a critical virulence factor of the plague pathogen. The manuscript covers all functions of this protease, including evolution, structural characteristics, enzymatic and adhesive activities, role in virulence, as well as practical applications for development of vaccines and diagnostics. The review is well written, logically structured, and contains recent references. This work will be of interest to general reader.

The reviewer has no major concerns on this study, and a few minor suggestions to improve the manuscript are listed below.

p.1, line 24. Specify this key inhibitor of the coagulation cascade. Generally, the proposed mechanism of the involvement of Pla in induction of coagulation should be described in more details (p. 7, lines 230-234)

p.2, lines 71-75. Mention that Pla has no cysteine residues, which is important for the structure

p.14, last page of Table 1. Add one more Y. pestis protein, such as KatY as those targeted by Pla, see PMID: 10322012 

p.22. Make a summary of the review and identify gaps of knowledge for the future research

Reviewer 2 Report

The review article by Sebbane et al. is an incredibly through and creative review article about the plasminogen activator of Yersinia pestis.  I found it to be a great resource and I know it will be received well by the scientific community.

Minor Issues

Line 24 change to “inactivates”

Line 62 and throughout avoid “stress out” this implies a stressed psychological state…I would simply say “we would like to stress here..”  “We would like to emphasize here…”  We would like to point out here…”

Line 129 change to “neither at the transcriptional nor at the translational level”

Line 157: spell out LOS first time used

Line 210: I am not sure what is implied with the work “fabulous”  do you mean “complex”, “complicated”, “extensive”…I do not believe fabulous is the correct word choice.

Line 258.  As a child from the 1980s, I loved the pacman reference (just remove the K J)

Line 264:  avoid “stress out” see above

Line 270: change to Convincing evidence

Line 273: delete “of” in “ over a dozen of Pla targets”

Line 310: Change to different portions of the passenger domain of YapG and YapE are released…

Line 335: consider changing to read …which in addition to its role in the cleavage (activation) of plasminogen to generate plasmin, is also important…

Line 340:  I may have missed it…spell out IDPRs first time?

Line 355: delete “it occurs that”

Line 381: change to Y. pestis genes

Line 388: delete “or not”

Line 410: I enjoyed the comparison with a Swiss army knife, I would just quotations “Swiss army knife”

Line 447.  The sentence beginning “Therefore, whether Y. pestis…” is awkward, consider revising.

Line 463:  what are 70-90% and 10-30% referring to? It was unclear to me.

Line 488: I believe you should delete “not” …Pla is dispensable

Line 519:  Data is plural…change to “The later data are”
